# Impact of GLP-1 Agonists on Male Reproductive Health—A Narrative Review

**DOI:** 10.3390/medicina60010050

**Published:** 2023-12-27

**Authors:** Alexandra Aponte Varnum, Edoardo Pozzi, Nicholas Allen Deebel, Aymara Evans, Nathalie Eid, Hossein Sadeghi-Nejad, Ranjith Ramasamy

**Affiliations:** 1Desai Sethi Urology Institute, Miller School of Medicine, University of Miami, Miami, FL 33136, USA; aav142@miami.edu (A.A.V.); exp1710@med.miami.edu (E.P.); a.evans5@med.miami.edu (A.E.); 2Department of Urology, University Vita-Salute San Raffaele, 20132 Milan, Italy; 3Division of Experimental Oncology, Unit of Urology, Urological Research Institute, IRCCS Ospedale San Raffaele, 20132 Milan, Italy; 4Department of Urology, Wake Forest University School of Medicine, Winston-Salem, NC 27101, USA; ndeebel@wakehealth.edu; 5Grossman School of Medicine, New York, NY 10016, USA; nathalie.eid@nyulangone.org (N.E.); hossein.sadeghi@nyulangone.org (H.S.-N.)

**Keywords:** glucagon-like peptide-1 receptor agonists, obesity, semen parameters, male infertility

## Abstract

*Background and objective*—Obesity is a prevalent health concern that notably impairs male fertility through hormonal disruptions and other pathophysiological alterations. Glucagon-like peptide-1 receptor agonists (GLP-1 RAs) can significantly reduce weight. This narrative review synthesizes the existing literature discussing the impact of glucagon-like peptide-GLP-1 RAs on the male reproductive system, particularly on the hypothalamic–pituitary–gonadal axis and spermatogenesis, highlighting their potential impact on male fertility. *Material and methods*—PubMed database was used for the retrieval of English-language articles published up to November 2023. This non-systematic literature review predominantly concentrates on both pre-clinical and clinical studies pertaining to GLP-1 RAs, specifically exploring their impact on male reproductive hormones and sperm parameters. *Results*—GLP-1 receptors have been identified within the male reproductive system according to the existing literature. While the exact mechanisms are not well understood, they appear to be involved in glucose homeostasis and energy metabolism, both vital processes in spermatogenesis. Multiple clinical trials have demonstrated the efficacy of GLP-1 RAs for promoting weight loss. Recent studies show that the use of GLP-1 RAs in obese males may enhance sperm metabolism, motility, and insulin secretion in vitro, along with positive effects on the human Sertoli cells. Recent clinical trials discussed in this review demonstrate weight loss associated with GLP-1 RAs is correlated with improvements in sperm count, concentration, and motility. However, the direct impact of GLP-1 RAs on male reproductive hormones remains unclear, necessitating further research to confirm their potential role in treating male infertility. *Conclusions*—This narrative review summarizes the existing literature discussing the potential impact of GLP-1 RA on the male reproductive system, emphasizing their potential therapeutic role in addressing idiopathic infertility in obese men. Despite numerous studies exploring the influence of GLP-1 and GLP-1 RAs on reproductive hormones, testicular function, and spermatogenesis, further clinical trials are crucial to validate initial evidence. Longer follow-up periods are essential to address uncertainties regarding the long-term repercussions and outcomes of GLP-1 RA use. While this holds true, the current literature suggests that GLP-1RAs show promise as a potential therapeutic approach for improving sperm parameters in obese men.

## 1. Introduction

Obesity is a pervasive and complex disease that has become an epidemic affecting more than 650 million adults [1,2]. Obesity is characterized by abnormal or excessive accumulation of fat and is associated with many medical conditions, including cardiovascular diseases, diabetes mellitus, hypertension, osteoarthritis, mental illnesses, and several cancers [3,4,5]. Furthermore, it has been correlated to significant clinical, psychosocial, and economic burdens as a result of increased morbidity and mortality [2,6,7]. In males of reproductive age, obesity is also an important risk factor for male factor infertility (MFI) and has been found associated with worsened assisted reproductive technology (ART) outcomes [8,9,10]. Several population-based studies have demonstrated an association between Body Mass Index (BMI), MFI, and subfecundity, defined as time to pregnancy or lack of conception after ≥12 months of unprotected intercourse [8]. Meta-analyses have also concluded that higher BMIs in males correlated with a decrease in fertility and in vitro fertilization (IVF) success [10,11].

The mechanisms by which obesity impairs male reproductive capabilities are multifactorial and interrelated. Excess body fat may influence sperm production and quality, hormones involved in the male reproductive system, and erectile function [12,13]. In this regard, many studies have shown these associations [8,14,15,16]. In a systematic review, Serenade et al. analyzed 21 studies and concluded that obese males were more likely to have oligospermia and azoospermia [11]. Ma et al. similarly showed that obese males had a reduction in sperm quality, including semen volume, sperm number, and total motile sperm count [14], but these findings are inconsistent across studies [8,15,16]. While the effect of obesity on sperm parameters has been less consistent, its effects on hormones are well elucidated.

Male obesity can dysregulate hormones in the hypothalamus–pituitary–gonadal axis (HPGA), causing secondary hypogonadism [15,17,18]. Hypogonadism is defined as a deficient or absent gonadal function that results in insufficient testosterone production [19]. Physiologically, the hypothalamus releases gonadotropin-releasing hormone (GnRH), prompting the pituitary gland to release luteinizing hormone (LH) and follicle-stimulating hormone (FSH), which stimulate testosterone production and spermatogenesis in males [20]. Excess adipose tissue leads to increased peripheral aromatization of testosterone, which results in higher levels of estradiol [15,21]. Estradiol is more biologically active than testosterone and its high levels in obese males inhibit the HPGA via negative feedback, impairing testosterone synthesis and spermatogenesis [20]. Furthermore, studies have demonstrated that hyperinsulinemia associated with higher BMIs leads to reduced production of Sex Hormone-binding Globulin (SHBG), which normally inhibits the biological activity of sex hormones, amplifying the effects of increased estrogen [15,17,20,22]. Male obesity is also associated with lower levels of inhibin B, a growth-like factor that stimulates testosterone secretion in Sertoli cells. Lastly, excessive adipose tissue triggers the secretion of pro-inflammatory cytokines and adipokines, which further inhibit testosterone production [20]. While few studies have evaluated the direct effect of weight loss in obese males on fertility, some have demonstrated that weight loss in this group is associated with rises in testosterone and improved sperm parameters [18,23]. Current interventions for weight loss include lifestyle modifications and surgical interventions. Glucagon-like peptide-1 (GLP-1) receptor agonists, a class of medications used for type 2 diabetes and obesity, present a new avenue for MFI management in obese men [24]. The present narrative review aims to comprehensively assess existing research on the impact of GLP-1 receptor agonists (GLP-1 RA) on MFI and shed light on the prospective utility of these medications within this specific population.

## 2. Methods

Online databases such as PubMed, Google Scholar, and Science Direct were utilized to ensure a comprehensive exploration of the relevant literature. Key search terms included “GLP-1 receptor agonists”, “male infertility”, “BMI and male infertility”, GLP-1 RAs and weight loss”, and “GLP-1 RA and male reproductive system”. The selection process involved extensively reviewing the literature to select those relevant to the subject matter, with emphasis on the most recent clinical trials pertaining to weight loss, GLP-1 RAs, and male fertility.

## 3. Glucagon-like Peptide-1 Receptor Agonists (GLP-1 RAs)

Glucagon-like peptide-1 receptor agonists (GLP-1 RA) are promising agents in the treatment of obesity with potential applications to MFI. These agents exert their therapeutic effects through the activation of GLP-1 receptors present in various tissues, including the intestine, heart, kidney, brain, immune cells, and pancreatic islets. Each of these tissues responds to GLP-1 RA with distinct pharmacological actions and encompasses a wide range of metabolic and physiological processes [25]. GLP-1 RAs in the intestine are stimulated and lead to increased growth and decreased lipoprotein secretion [25]. In the heart, they enhance glucose utilization, cardiac function, and vasoprotection, while reducing fatty acid metabolism and inflammation [25]. The kidney experiences increased sodium excretion, and the brain benefits from neuroprotection, neurogenesis, and decreased food intake [25]. Additionally, immune cells exhibit a decrease in inflammation, and pancreatic islet cells witness an increase in insulin secretion, biosynthesis, proliferation, and somatostatin secretion. In contrast, apoptosis of Beta cells and glucagon secretion is decreased [25]. Clinically approved and investigational GLP-1 agonists, such as albiglutide, dulaglutide, exenatide, exenatide extended release, liraglutide, semaglutide, lixisenatide, and efpeglenatide, can be categorized as short acting (e.g., lixisenatide and exenatide) with significant fluctuations in plasma levels and a half-life of 2 to 5 h, or long-acting (e.g., semaglutide, dulaglutide, exenatide extended release, liraglutide, albiglutide, and efpeglenatide), which consistently activate GLP-1 receptors with a longer half-life (12 h to several days) as illustrated in Table 1 [26,27]. Short-acting agonists impact fasting blood glucose levels, postprandial hyperglycemia, insulin secretion, glucagon secretion, and gastric emptying, while long-acting agonists exhibit stronger effects on these parameters [26,27].

With the potential role of GLP-1 RA in weight management and male infertility, it is important to also highlight the adverse effects associated with the administration of this drug. Gastrointestinal symptoms, including nausea, vomiting, and diarrhea, are the most commonly reported side effects [28]. Up to 50% of patients treated with GLP-1 RA may experience nausea [27,28]. Short-acting GLP-1 RA induces nausea that attenuates slowly over weeks to months, while long-acting GLP-1 RA leads to quicker attenuation within 4 to 8 weeks [28]. It is worth noting that while nausea is common, it is generally mild and tends to lessen over time with continued treatment. Slower titration to the maximum effective dose has been shown to decrease the incidence of gastrointestinal side effects. The risk of acute kidney injury has also been discussed, potentially explained by volume contraction secondary to gastrointestinal losses [29]. Within the major clinical trials investigating the use of GLP-1 RA for weight loss, gallbladder-related adverse events ranged from 0.8% to 3.1% in the liraglutide treatment group of the STEP 8 trial [30]. Across the SCALE, STEP 1, and STEP 8 clinical trials, the incidence of acute pancreatitis was below 1% [30,31,32,33]. While the use of GLP-1 RAs alone is associated with a low incidence of hypoglycemia, there may be a greater risk when combined with other diabetic medications such as insulin and sulphonyurea [29]. Notably, the current literature has not revealed any discernable adverse effects on the male reproductive system.

### GLP-1 RAs and Obesity

In recent years, there have been several clinical trials investigating the use of GLP-1 RAs in chronic weight management. The SCALE trial focused on patients with type 2 diabetes, treating them with either 1.8 mg or 3.0 mg of liraglutide. After 56 weeks, those receiving the higher dose achieved 6% weight loss compared to 4.7% in the lower dose group and 2% in the placebo group [31]. In a Phase 3a component of the SCALE trial, liraglutide was investigated in obese individuals without diabetes. After 56 weeks, the treatment group saw an average weight reduction of 8.4 kg compared to 2.8 kg in the placebo group, with 33% of liraglutide-treated patients losing more than 10% of their body weight [32]. Similar results were seen in the pivotal STEP 1 trial, where semaglutide resulted in an average weight reduction of 14.9% compared to 2.4% in the placebo group [33]. Additionally, Rubino et al. compared semaglutide and liraglutide effects on weight loss in individuals without diabetes in the STEP 8 trial. At the completion of the 68-week treatment plan, investigators found weight loss to be superior in the group that received semaglutide once weekly, with a mean weight change of −15.8% in this group [30]. More recently, the SURMOUNT trial has investigated the potential for tirzepatide as another GLP-1 RA drug to treat individuals with obesity [34]. The results of this trial demonstrated that tirzepatide led to a mean weight reduction of −20.9% in the group receiving 15 mg weekly [34].

The mounting evidence supporting the weight-reducing effects of GLP-1RA has sparked interest in exploring their potential impact on male factor infertility (MFI). Obesity, a widespread health concern, significantly hampers male fertility by causing hormonal disruptions and other pathophysiological changes. Hormonal imbalances stemming from alterations in the hypothalamic–pituitary–gonadal axis play a crucial role in how excess weight adversely affects semen parameters. Additionally, the accumulation of visceral fat leads to elevated estrogen levels and diminished testosterone levels, disrupting sperm production and quality. Hence, the exploration of GLP-1RA for chronic weight management stands out as a prominent area of research in the current landscape of obesity and MFI studies. A noteworthy consideration in the current discourse surrounding GLP-1 RAs is the escalating cost associated with this medication. The heightened demand has led to a significant increase in out-of-pocket costs for patients, which is a factor that warrants careful consideration when discussing the use of GLP-1 RAs with patients. Concurrently, within the realm of MFI treatments, costs are often also substantial with minimal insurance coverage available within the United States [35]. It is imperative to acknowledge the potential financial burden associated with GLP-1 RAs, as it could pose a significant barrier to treatment for many individuals.

Indeed, studies have shown that individuals undergoing bariatric surgery often experience a substantial 25–30% weight reduction within 1–2 years post surgery [36]. In this regard, in a retrospective study of 907 men, including 109 men with azoospermia, Ajayi et al. found that obese men (BMI > 30) were nearly twice as likely to have azoospermia [37]. This finding is similar to a metanalysis performed by Semondade et al., that included 9779 men in which authors found a relative risk of azoospermia 1.8 among obese men [36,38]. Given the significant weight loss potential associated with GLP-1 RA use, there is a growing preference for a less invasive therapeutic alternative to surgery, making it a more appealing option for obese patients dealing with MFI.

In conclusion, further research is imperative in exploring the relationship between GLP-1 RA, obesity in men, and its impact on MFI.

## 4. GLP-1 RAs and Male Reproductive Physiology: A Comprehensive Examination

### 4.1. GLP-1 RAs and Male Infertility

While GLP-1 RAs have a clear impact on weight loss, which consequently improves hormonal regulation, it has also been recently elucidated that GLP-1 RA may exert additional effects at the testicular level (Table 2). Recent findings suggest that apart from their well-documented impact on the intestine and pancreas, such as delaying gastric emptying and stimulating insulin secretion to facilitate weight loss and enhance glycemic control, GLP-1Rs may also exert additional effects at the testicular level [39,40]. Through several studies, the presence of GLP-1 receptors in Sertoli and Leydig cells has been established [41,42,43,44]. Caltabiano et al. utilized immunohistochemistry to identify these receptors in normal and neoplastic human testicular tissues, as well as rodent testes [44]. Their study proposes that GLP-1 receptors play a role in onco-suppression, Leydig cell function, and hormone secretion and function [44]. GLP-1 has also been recognized as a regulator of glucose homeostasis and energy metabolism, which are vital processes in spermatogenesis [43]. Martins et al. conducted a study investigating the effects of GLP-1 in human Sertoli cells as they relate to metabolism and mitochondrial function [43]. In this study, human Sertoli cells were cultured in the presence of increasing GLP-1 doses, spanning physiological post-prandial levels to those observed in pharmacological treatment [43]. At the highest GLP-1 concentrations, authors observed a decrease in mitochondrial membrane potential and oxidative damage [43]. At the lowest GLP-1 concentrations, improved efficiency in the conversion of glucose to lactate was seen [43]. This study ultimately demonstrated the presence of GLP-1 RA’s in Sertoli cells and identified their role in testicular energy homeostasis. Rago et al. identified analogous functions of GLP-1 by discerning GLP-1 receptors in sperm cells [42]. Their study revealed sperm as a specific target of GLP-1 incretin. To represent native GLP-1, the employed exendin-4 due to its structural similarities. Using exendin-4 in a dose–response model, they determined that GLP-1 and GLP-1 receptors are implicated in mediating progressive motility and cholesterol efflux, with an associated increase in kinase pathway signaling [42]. Stimulation of glucose metabolism enzyme activities was also seen, underscoring the role of GLP-1/GLP-1R in glucose metabolism and cellular activities within the testicles [42]. Studies exploring the interplay between GLP-1 RA, weight loss dynamics, and semen parameters have been conducted utilizing mice and rat models. In obese mice, they observed increased mRNA expression of TNF-α, MCP-1, and F4/80 within the testis, thereby exerting a negative effect on sperm quality [45]. Administration of exenatide, a GLP-1 RA, daily for 8 weeks in obese mice resulted in improved sperm motility, DNA integrity, and decreased expression of pro-inflammatory cytokines [45]. The current approach to managing secondary hypogonadism in obese men with oligospermia commonly involves the use of clomiphene citrate and anastrozole. Clomiphene citrate, a selective estrogen receptor modulator (SERM), acts as an estrogen antagonist to enhance LH and FSH secretion. Currently used off label, clomiphene citrate effectively raises testosterone levels and improves sperm concentration in this specific patient population [46,47,48]. Anastrozole, an aromatase inhibitor, prevents the peripheral conversion of androgens to estradiol [49]. Anastrozole, a selective aromatase inhibitor, has traditionally been recommended for men with testosterone-to-estrogen (T:E) ratios below 10 [49]. While the American Urological Association guidelines endorse the use of aromatase inhibitors for individuals with low testosterone, current support for treating idiopathic infertility in men primarily revolves around follicle-stimulating hormone analogs [49]. Although both are viable treatments for enhancing hormonal balance and semen parameters, clomiphene citrate has rare but documented side effects, including thromboembolism, gastrointestinal distress, and occasional weight gain in men [50]. Furthermore, despite clomiphene citrate’s association with significant increases in sperm concentration, it is not universally effective, with a meta-analysis indicating a significant increase in sperm concentration in approximately 60% of men [51]. For men facing the dual challenge of normal sperm count and low testosterone, who seek to preserve fertility, Natesto presents itself as a viable option [52,53]. However, it is noteworthy that even with Natesto, there exists a small risk of negatively impacting sperm concentration [52,53]. While identifying treatment gaps within this patient demographic, it becomes apparent that obese men, characterized by hypogonadism who wish to preserve fertility, could represent an ideal group for potential intervention with GLP-1 RAs. In instances where obese males exhibit oligospermia alongside normal testosterone and estradiol concentrations, conventional pharmaceutical approaches like clomiphene may not be suitable. Studies indicate that around 45% of obese males may present with subnormal testosterone levels, yet a substantial proportion falls within the broad spectrum of normal testosterone levels [54]. Within this cohort, there also exists an opportunity for pharmacological intervention with a GLP-1 RA. Examining the outcomes of in vitro fertilization (IVF) in obese men reveals a discernible negative impact when compared to men with a normal BMI. This discrepancy is potentially attributed to diminished sperm quality, as highlighted by a recent study [55]. Lower fertilization rates and the presence of overall lesser-quality embryos in individuals with a BMI exceeding 28 kg/m^2^ were identified. Additionally, the study uncovered a noteworthy disparity in sperm telomere length, with obese men exhibiting shorter telomeres compared to the normal BMI group [55]. This observed difference in telomere length likely contributes to poor sperm quality and overall inferior IVF outcomes in this demographic. As a result, couples with obese men experienced a reduced clinical pregnancy rate [55]. This underscores the importance of addressing the specific needs of this patient population. Even with sufficient sperm concentration, it is crucial to prioritize the enhancement of sperm quality, ensuring more favorable outcomes for obese men undergoing IVF.

#### GLP-1 RAs and Hypogonadism

While several prominent clinical trials have explored the use of GLP-1 RA for weight management, few have delved into their impact on hypogonadism. The correlation between obesity and hormonal dysregulation has been well studied, revealing that obesity can precipitate secondary hypogonadism due to hormonal dysregulation in the hypothalamus–pituitary–gonadal axis [13,15,18]. Numerous studies have consistently demonstrated an association between weight loss and increases in testosterone and sex hormone-binding globulin, with a concurrent decrease in estradiol levels [64]. Notably, this effect appears to be most pronounced in young men without diabetes [64]. In a randomized control trial investigating the impact of GLP-1 RA on the male reproductive axis, 18 healthy men with a mean BMI within the normal range were enrolled and administered a GLP-1 infusion (rate of 0.8 pmol/kg/min) for 500 min with serum sampling every 10 min [56]. There was no discernable effect on LH pulsatility, nor alterations in LH, FSH, or testosterone [56]. However, with a lack of obese participants enrolled in this study, it is difficult to make associations between GLP-1 and hormonal alterations in the male reproductive axis [56]. In another study, conducted by Jeibmann et al., the researchers investigated the impact of GLP-1 RA on LH and testosterone secretion in 9 healthy male volunteers. The subjects underwent an oral glucose tolerance test to assess LH, testosterone, and GLP-1 responses to glucose. Additionally, euglycemic clamp experiments were conducted with either saline or a constant infusion of GLP-1. The results showed that oral glucose ingestion led to a reduction in plasma testosterone levels at 30 min compared to baseline, despite unaltered LH levels [65]. On the other hand, constant GLP-1 infusion had no significant effect on LH, testosterone, FSH, and leptin levels [65]. Of note, pulse analysis revealed a decrease in the number of testosterone pulses and a tendency for increased pulse duration. Importantly, these findings suggest that both oral glucose ingestion and intravenous GLP-1 infusion reduce the pulsatile component of testosterone secretion independently of LH release [65]. In addition to the link between obesity and hormonal dysregulation, some of the literature has explored the impact of obesity on sperm epigenetics, with investigations suggesting that obesity may influence sperm microRNA content and DNA methylation patterns, despite many overweight or obese men being able to achieve natural pregnancies [8]. Given the vital role of these hormones in spermatogenesis, along with elevated peripheral aromatization of testosterone in the presence of excess adipose tissue, it is unsurprising that several studies have found obese males to have an increased likelihood of demonstrating oligospermia [11,14]. The preliminary findings of a recent clinical trial unveiled the impacts of liraglutide, on treating obese, fertile men with metabolic hypogonadism [57]. In this study, 110 male patients between 18–35 years old with metabolic hypogonadism were placed in three groups as per their current fertility status: Group A with patients desiring fertility, Group B representing patients with no desire for fertility, and Group C comprised those who previously had a child. Patients within Group B were treated with liraglutide, while participants in Group A received urofollitropin and those in Group C were given transdermal testosterone [57]. Treatment spanned a total of four months. Results showed a significant improvement in semen parameters (sperm concentration, sperm motility, morphology) within the group treated with liraglutide (Group B), in comparison to baseline metrics and Group A, in particular sperm motility [57]. Semen analysis was not performed on Group C participants. Furthermore, erectile function was significantly improved in Group B, compared to Groups A and C [57]. Participants treated with liraglutide exhibited markedly higher total serum testosterone levels than those who did not receive this treatment [57]. Based on these preliminary findings, liraglutide holds the potential for efficacious treatment of metabolic hypogonadism among obese males [57]. In a study by Jensterle et al., the efficacy of liraglutide in addressing obesity-associated functional hypogonadism in men was investigated. The study compared the impact of liraglutide with testosterone replacement therapy (TRT) on functional hypogonadism. Participants were assigned to receive either liraglutide (3.0 mg daily) or TRT in the form of daily transdermal gel, with a short-term treatment duration of 16 weeks. Total testosterone increased significantly in both the TRT and liraglutide groups, with no statistically significant difference between the two treatment groups (+5.9 nmol/L in TRT versus +2.6 nmol/L in liraglutide group) [58]. Among those receiving TRT, LH and FSH were further suppressed, as expected, while an increase in LH and FSH was observed in the liraglutide group, contributing to the recovery of the hypothalamus–pituitary–testicular HPT axis [58]. Participants receiving liraglutide experienced a greater anticipated weight loss (mean 6% weight reduction), contrasting with the minimal weight loss in the TRT group [58]. Both groups exhibited improvements in libido and sexual function, as measured by morning erections and ejaculations per week, with no significant differences observed between the treatment groups [58]. Giagulli et al. similarly found that the addition of liraglutide to a treatment regimen involving TRT and metformin enabled men with overt hypogonadism and erectile dysfunction (ED) to reach testosterone levels within the range of 300 ng/dL, accompanied by improvements in ED and glycemic control [59]. Both studies reported overall improved glycemic control in participants receiving liraglutide [58,59]. Jensterle et al. also highlighted the correlation between the degree of weight loss and the magnitude of testosterone increase, aligning with previous studies that established a relationship between weight loss and total testosterone levels [58,64]. These studies underscore the potential of liraglutide as a treatment option for hypogonadism, especially in patients who may also benefit from improved glycemic control [58].

### 4.2. GLP-1 RAs and Semen Parameters

While prior investigations have found no significant association between BMI and semen parameters, more contemporary data refute these observations [63,66,67]. This suggests that our understanding of the relationship between weight and male reproductive health is complex and multifactorial. Guo et al. conducted a comprehensive review discussing the decline in sperm count and concentration associated with a high BMI [60]. This observation is supported by multiple studies, with sperm concentration being the most commonly associated sperm parameter affected [11,23]. In a cohort study by Hakonsen et al., 43 men were observed over 14 weeks while they were enrolled in a residential weight loss program [23]. At the start of the study, an inverse correlation was noted between BMI (ranging from 33 to 61 kg/m^2^), total sperm count, sperm concentration and motile sperm [23]. Approximately 34% of men in this study exhibited oligospermia (<15 million/mL) [23]. Following 14 weeks, the median weight reduction was 22 kg (median of 15% weight loss), and a positive correlation between weight loss and an increase in total sperm count was observed [23]. Faure et al. further investigated this concept by examining sperm DNA fragmentation improvements with weight loss in a small study involving six patients undergoing lifestyle modifications [61]. They observed a decrease in sperm DNA fragmentation at the end of the study along with a significant increase in the testosterone/oestradiol ratio [61]. However, they did not find a significant change in other semen parameters, potentially attributed to the small sample size [61]. It is pertinent to note that among the six patients, only two had a BMI above 30 [61]. Mir et al. additionally found weight loss to be associated with a statistically significant improvement in sperm DNA fragmentation index (DFI) and sperm morphology, with a mean DFI of approximately 20% prior to weight loss, and 17.5% at the end of the study [62]. Approximately half of the participants had DFI <20% at the start of the study, with 44% in the 20–40% range, and 2.9% greater than 40% [62]. In a singular case report in a patient with idiopathic infertility and initially normal semen parameters, a notable transition to severe oligospermia and later azoospermia occurred after a 4-month period. The patient reported no changes to lifestyle or medical history, with 0.6 mg liraglutide being the only reported medication which was started one month prior to the first normal semen analysis [68]. During the 5 months of being on liraglutide, he lost 2 kg [68]. Similar findings have not been reported in any of the other literature to our knowledge. On the other hand, in a more recent study on weight loss and improvements in sperm parameters, Andersen et al. conducted a randomized control trial as a sub-study of the S-LiTE trial. They assigned 56 men to receive either liraglutide or a placebo, with or without an adjunct exercise program. Before entering these four groups, all participants underwent an 8-week low-calorie diet program. After the 8-week period, weight loss and sperm parameters were evaluated. In the beginning, the mean decrease in BMI was 5, with an initial average BMI of 37 [63]. At the start, 17% of participants had oligospermia which decreased to 13% after the 8-week intervention, further supporting the link between weight loss and improved sperm parameters [63]. Specifically, there was a 1.71-fold increase in sperm concentration and a 1.41-fold increase in sperm count after 8 weeks [63]. This improvement was maintained after one year, but only in individuals who maintained at least a 12 kg weight loss, regardless of their treatment group. Sperm motility and volume remained unchanged, and there was a non-statistically significant increase in motile sperm [63]. Beyond the initial 8-week diet intervention, liraglutide did not have additional effects on semen parameters [63]. However, the study suggested the potential use of liraglutide to help maintain weight loss and, in turn, preserve sperm quality [63]. In conclusion, current evidence suggests that using GLP-1RA in preconception treatment may offer new approaches for managing weight and infertility in men with obesity. However, a comprehensive understanding of the intricate connections between metabolism and reproduction is essential. Monitoring the hormonal profile of the HPG axis, along with repeated semen analyses including in-depth assessments of sperm quality, such as sperm DNA integrity and epigenetics, is necessary. Furthermore, it is crucial for subsequent studies to explore the efficacy of various GLP-1 RAs in the treatment of MFI, as the current literature has not yet explored this. These questions lay the foundation for future studies exploring the implementation of GLP-1 RA in assisted reproduction and reproductive health management for men with obesity [69].

## 5. Conclusions

This comprehensive narrative review synthesizes the existing literature discussing the impact of GLP-1 RAs on the male reproductive system, highlighting their potential as a therapeutic intervention for idiopathic infertility in obese men. Although numerous studies have explored the influence of GLP-1 and GLP-1 RAs on the reproductive hormonal axis, testicular function, and spermatogenesis, further clinical trials recruiting infertile obese men are imperative to substantiate these effects. Additionally, longer follow-up time is essential, given the ambiguity surrounding the long-term repercussions and outcomes of GLP-1 RA use. Furthermore, it is imperative for future investigations to document and analyze both pregnancy rates and live birth rates, aspects frequently omitted from the current literature. While further studies are necessary, the current literature elucidates that GLP-1Ras emerge as a potentially promising therapeutic avenue for improving semen parameters in obese men.

## Figures and Tables

**Table 1 medicina-60-00050-t001:** Available GLP-1 RAs, their duration of action, half-life, and common dosing.

GLP-1 Agonist	Half Life	Common Dosing
**Short-Acting**		
Exenatide	~2–5 h	5 mcg up to 10 mcg with titration twice daily
Lixisenatide	~2–5 h	10 mcg up to 20 mcg with titration once daily
**Long-Acting**		
Exenatide ER	~12 h to several days	2 mg once weekly
Liraglutide	~13 h	0.6 mg up to 1.8 mg with titration once weekly
Dulaglutide	~5 days	0.75 mg up to 4.5 mg with titration once weekly
Semaglutide (subcutaneous)	~7 days	0.25 mg up to 2 mg with titration once weekly
Semaglutide (oral)	~7 days	3 mg up to 14 mg with titration once daily
Albiglutide	~5–8 days	30 mg up to 50 mg once weekly
Efpeglenatide	~5–8 days	4 mg or 6 mg once weekly 8 mg, 12 mg, or 16 mg once monthly
Tirzepatide (dual-acting GLP-1 and GIP agonist)	~5 days	2.5 mg up to 15 mg with titration once weekly

**Table 2 medicina-60-00050-t002:** Pre-clinical and clinical studies investigating the role of GLP-1 RAs on semen parameters.

Study	Study Design	Participants	Intervention	Main Findings	Adverse Events
Martins et al. (2019) [43]	Pre-clinical	NA	Increasing GLP-1 doses in the presence of human Sertoli cells	* Decrease in mitochondrial membrane potential and oxidative damage at highest GLP-1 dose * Improved efficiency in conversion of glucose to lactate at lowest GLP-1 dose	NA
Rago et al. (2020) [42]	Pre-clinical	NA	Dose-response model using exendin-4 to investigate GLP-1 receptors on human sperm cells	* GLP-1 and its receptors play a role in mediating progressive motility and cholesterol efflux * Observed stimulation of glucose metabolism enzyme activities	NA
Zhang et al. (2015) [45]	Pre-clinical (Mouse model)	NA	12 weeks of chow or high fat diet (HFD) followed by HFD mice assigned to saline or exenatide daily for 8 weeks	* Improved sperm motility, DNA integrity, decreased expression of pro-inflammatory cytokines	NA
Izzi-Engbeava C et al. (2020) [56]	Randomized clinical trial	18 men (normal range BMI, eugonadal)	GLP-1 infusion (rate 0.8 pmol/kg/min) for 500 min, serum sampling every 10 min	* No discernable effect on LH pulsatility, FSH, LH or testosterone levels	Nausea
La Vignera S. et al. (2023) [57]	Randomized clinical trial	110 male, aged 18–35, metabolic hypogonadism	Urofollitropin (Group A), liraglutide (Group B), transdermal testosterone (Group C) for 4 months of treatment	* Improvement in all sperm parameters and erectile function in Group B (sperm motility in particular) * Increased total serum testosterone and sex hormone-binding globulin in Group B * Significantly higher gonadotropin levels in Group B compared to other groups	NA
Jensterle et al. (2019) [58]	Prospective randomized open-label	30 obese men	Liraglutide (3.0 mg daily) or daily transdermal testosterone gel (TRT) for 16 weeks	* Increase in total testosterone and sexual function with no significant difference between treatment groups * Suppression of LH and FSH in TRT group, with increase in these hormones in liraglutide group * Greater weight loss in liraglutide group compared to TRT	Mild to moderate, transient gastrointestinal distress, none in TRT group
Giagulli et al. (2015) [59]	Retrospective observational	43 obese, diabetic, hypogonadal men	Testosterone undecanoate (TU) and metformin for 1 year, with liraglutide added to poor responders for 1 year, and good responders maintained TRT and metformin	* Improvement in International Index of Erectile Function (IIEF) in group that received liraglutide versus no change in TRT/metformin (Met) group * TRT and Met after 2 years saw statistically significant rise in A1c and weight compared to year 1 * Greater glycemic control in group that received liraglutide	Transient gastrointestinal distress
Hakonsen et al. (2011) [23]	Prospective cohort study	43 obese men	14-week weight loss program	* Decrease in BMI associated with increase in sperm concentration, total sperm count, sperm morphology * Statistically significant increase in total sperm count and normal sperm morphology in the group with largest weight reduction	None
Guo et al. (2017) [60]	Systematic review and meta-analysis	NA	NA	* Decline in total sperm count, sperm concentration, and semen volume with increasing BMI	NA
Faure et al. (2014) [61]	Case series	6 men in infertile couples	Lifestyle modifications	* Significant improvement in sperm DNA integrity with weight loss, regardless of BMI * Significant increase in testosterone/oestradiol ratio * No significant change in other semen parameters	None
Mir et al. (2018) [62]	Cross-sectional	NA	Lifestyle modifications	* Statistically significant improvement in sperm DNA fragmentation index and sperm morphology with weight loss	None
Andersen et al. (2022) [63]	Randomized clinical trial	56 men	Liraglutide or placebo, with or without adjunct exercise program 8 week low calorie diet program for all participants prior to being assigned to group	* Improvement in sperm concentration and sperm count after 8 week low-calorie diet, maintained after 1 year individuals who maintained at least 12 kg weight loss regardless of treatment group * Sperm motility and volume unchanged * Liraglutide did not have any additional effects on semen parameters, however the most sustained improvement in sperm parameters was within the liraglutide group	None

Keys. NA: Not Available; BMI: Body Mass Index; TRT: Testosterone Replacement Therapy; GLP-1: Glucagon-like Peptide-1; FSH: Follicle-stimulating Hormone, Luteinizing Hormone.

## Data Availability

No new data were created or analyzed in this study. Data sharing is not applicable to this article.

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
