# Peer review of "Impact of GLP-1 Agonists on Male Reproductive Health—A Narrative Review"

_medicina, 2023, doi:10.3390/medicina60010050_

Round 1
Reviewer 1 Report
Comments and Suggestions for Authors
Authors conducted review about the impact of GLP-1 agonists on male reproductive health.
In table 1 authors already divided all drugs into 2 groups and there is no need for column Duration of action. Table 1 has low quality. Explain the reason of choosing exact trade names. Some of the drugs has several trade names, and what about multicomponent drugs containing GLP-1 RA?
Authors provide a discussion on adverse effects, especially gastrointestinal symptoms, is essential for a balanced view of the potential drawbacks of GLP-1 RA administration. Besides gastrointestinal symptoms, are there any other notable adverse effects associated with GLP-1 RAs that could be relevant to male fertility? Understanding the safety profile is crucial for evaluating the overall feasibility of these agents in managing male factor infertility.
In the context of male factor infertility, are there specific GLP-1 RAs that have shown more promising results or fewer side effects in clinical studies? It would be interesting to know if there are variations in their effects on male reproductive health.
Please illustrate the mechanisms of action of GLP-1 RAs in the male reproductive system, highlight the pathways involved in hormonal regulation, testicular function, and spermatogenesis.
Discuss more about potential areas for future research in the field. What about cost implications of GLP-1 RA treatment compared to other interventions?
While your review offers valuable information on the impact of GLP-1 RAs on the male reproductive system, it appears that not all areas were thoroughly explored, and certain related issues were not extensively discussed.
Reviewer 2 Report
Comments and Suggestions for Authors
I’ve read with attention the narrative review by Ramasamy et al. that is interesting, well-organized, overall well-written and update. I've only some minor comments.
Abstract: it is a bit unbalanced, with results much shorter than the conclusions
Main text: Even if the review is narrative by nature, it would be nice if the authors could better detail how they decide to chose the references to be selected and discussed in the review. Table 1 should omit trade names while adding information that could be more useful than half-life such for instance the power of the listed drugs. Finally, the different impact of the listed drugs that could also improve fertility by acting on cardiometabolic risk factors levels should be also more clearly stressed.
Round 2
Reviewer 1 Report
Comments and Suggestions for Authors
I do not have more comments.